# Multi-Impacts of Spatial Self-Policing during COVID-19: Evidence from a Chinese University

**DOI:** 10.3390/ijerph191912172

**Published:** 2022-09-26

**Authors:** Yuan Sun, Zhu Wang, Zhi Qiu, Congyue Zhou

**Affiliations:** College of Civil Engineering and Architecture, Zhejiang University, Hangzhou 310058, China

**Keywords:** self-policing, spatial management, university campus, COVID-19, keyword frequency, online forum

## Abstract

Current research has focused on the impacts of the COVID-19 pandemic on university students’ physical and mental health conditions but has rarely examined the secondary effects caused by school management and prevention policies. Chinese universities generally took a self-policing strategy to address the COVID-19 pandemic. This study aimed to examine how the self-policing effect fluctuated during the pandemic, assessed from the perspective of university students. We collected monthly data from January 2020 to August 2022 from Zhejiang University’s online forum CC98 and analyzed the monthly frequency of keywords in the online posts’ titles. The dataset covered five topics: pandemic situations, epidemic prevention policies, campus access control, campus space use, and emotional conditions. The results showed that university students have expressed concern about the pandemic over the past thirty-two months, which still has an unignorable influence on their lives and studies. They paid more attention to the epidemic prevention policies, which directly affected their social connections, spatial use, and psychological well-being. University students gradually questioned their duty to obey and showed impatience and resistance toward school self-policing management, especially during the second Omicron wave. Additionally, the findings investigated an introverted trend for university students living in a gated campus environment. In conclusion, we call for reflections on the current Chinese campus self-policing strategy to cope with future long-term and normalized pandemic situations. The concerns of university students should be taken into account as we move toward a post-COVID-19 world.

## 1. Introduction

Nearly three years have passed since the COVID-19 outbreak in late 2019 [1]. Human cities and societies are gradually recovering from the worldwide pandemic threats and are making changes toward a post-COVID-19 world [2].

There are rising reflections on the benefits and costs of the various policies, actions, and strategies for preventing the outbreak and defeating the pandemic [3]. McFarlane et al. [4,5] raised the core notion of a “political crowd” to shift urban governing impacts during pandemics. He pointed out that high-density crowds that accelerated high-risk outbreaks were politically ordered to be under control. Some scholars traced the relationship between human well-being and shifting COVID-19 factors through questionnaires [6], social media data [7], interviews [8], etc. Some collected data from digital platforms to examine multiple impacts of the pandemic on people’s daily behaviors [9], mental health conditions [10], and social networks [11]. Special attention was paid to teens and youths on campus, whose psychological development is not entirely mature. Since the pandemic raised unpredictable challenges for education and campus activities, conventional modes of teaching, studying, and laboratory practices on campus were partially adjusted and even displaced according to pandemic restriction policies and rules [12]. 

This study collected data from Zhejiang University’s online forum, officially named “CC98” [13], to analyze university students’ continuing concerns about COVID-19 pandemic events and the relevant factors of epidemic prevention policies, campus space use, and emotional expressions. Our findings provided evidence from China that university students’ monthly keywords in online post titles had particular associations with campus self-policing management in preventing COVID-19.

## 2. Literature Review on Self-Policing in Campus Management during COVID-19

### 2.1. Self-Policing for COVID-19 Prevention

“Self-policing” refers to keeping normal orders among a group of people living in communities or connected neighborhoods [14]. As an informal social control capacity in the urban system, self-policing is conducted by engaged community members or the public to achieve particular targets, such as reducing crime [15]. The authority of community governors is intended to achieve community bottom-up acceptance.

Laufs et al. reviewed the shifting roles that police play in pandemics [16] to maintain government–community relations. After the pandemic outbreak in China, the government rapidly issued relevant laws and regulations on COVID-19 prevention [17], including strict punishments of fixed-term imprisonment, criminal detention, or fines. This required broad engagement from community forces to cooperate with official police and staff so rigid enforcement of preventive orders like stay-at-home restrictions could be conducted in the community neighborhoods. In general, social uncertainty rises as the crowd population increases, taken as a risk for political stability, public health, and safety by the authorities [18]. Thus, it is a political preference of the Chinese central government and local governors to conduct neighborhood management and social services in enclosed communities with a small crowd for precise control [19]. Consequently, a vast, complex Chinese society has been divided into multiple single units or cells for efficient pandemic prevention through self-policing. 

In this context, community self-policing forces authorized by official orders were strengthened for pandemic prevention, according to top-down governmental decentralization [20]. Since restrictions on social distance and mobility have been proven effective in pandemic prevention, self-policing functions as public spatial control and individual mobility restriction in China’s grass-roots level community lockdown conditions [21]. At first, a lockdown was taken as an emergency plan during the short-term outbreak wave. This led to a flow-up action of spatial self-policing, which continued from January 2020 until now. In terms of pandemic prevention, a community gate is not only a simple access channel but a symbol of little infection risk within the gated environment. A “pandemic prevention unit (cell)” in Chinese neighborhoods and urban communities is a physically gated area in an enclosure pattern [22]. The growing sovereign power of self-policing controls the inner space and the crowds by holding the gates. However, in terms of urban liberalism [23], such a self-policing measure for pandemic prevention has affected the balance between urban control and liberty.

### 2.2. Campus Lockdown Strategies in Chinese Universities

The university campus is a type of community for academic and living activities. Some studies have suggested that schools should be the last to close and the first to open [24]. In China, however, campuses were still regarded as weak places in response to pandemic threats, overlooking their potential for public functions. Campus lockdown [25] has been a self-policing management strategy for outbreak prevention in China since 2020, enforced by university officials and campus guards. It seems that the enclosed campus after the COVID-19 pandemic became an independent kingdom, where university officials have almost absolute control of campus access, facilities, services, and activities [26]. They appealed to students to keep social distance [27] and decrease travel frequency; however, they ultimately set campus boundaries between universities and society. A robust defense system was created through walls, fences, and gates to preserve the internal campus safety from external pandemic risk as an enclave. Currently, in some cities, students living on campus can still connect with the public, but their mobility is under strict limitations by campus preventive orders. 

University lockdown by self-policing has disrupted the daily campus life of students, teachers, and scholars but has provided limited physical and psychological support for them. In order to balance the relationship between campus public order and individual living order during COVID-19, individuals were considered to sacrifice invisible benefits for pandemic prevention [28]. 

### 2.3. Negative Impacts of Campus Self-Policing on University Students

#### 2.3.1. Impacts on the Social Aspects

Sun et al. [29] found that campus lockdown by self-policing minimized students’ social contacts, whose academic network, based on academic conferences, lectures, competitions, exhibitions, etc., was largely blocked. They failed to trigger multidiscipline innovation with other universities and corporations at home and abroad because academic mobility was severely restricted [30]. In addition, this decreased the opening of the campus to society; external public could not access the advanced educational resources [31].

#### 2.3.2. Impacts on Spatial Aspects 

The campus enclave leads to an increasingly heavy burden on spatial management and daily operation [32]. On some occasions, existing rooms and facilities on campus might not be able to provide sufficient activities and services for laboratory practice, entertainment, socializing, and internship. Some university students gradually become tired of changeless surroundings, with few opportunities to experience new interests from the external social environment [33]. Bennett et al. [34] detailed the dietary habit changes that resulted from campus lockdown.

#### 2.3.3. Impacts on Mental Aspects

Abdullah et al. [35] investigated two psychological factors (depression and stress) that led to lower quality of life among university students during pandemics. The enclosure of academic activities and social life on campus aggravated mental isolation among the young and guided an impulse for them to escape from the “prison” caged by the virus [36]. Tensions appeared in the relationship between students and university officials. Because of the reduction of social diversity, the campus environment was simplified, where school officials were easier to fixate on due to their dominant status and established ideologies. The intervention of campus guards led to students feeling ordered and controlled [37]. However, such top-down order for the youths potentially raised rebellious emotions and behaviors towards authorities [38]. 

## 3. Materials and Methods

### 3.1. Study Area

Zhejiang University in Hangzhou, China, was chosen as a suitable study case to validate the impacts of self-policing management in response to pandemic prevention. Hangzhou is the capital of Zhejiang Province, an essential megacity in the eastern developed areas of China. Zhejiang University is one of China’s most prominent universities, with five campuses in Hangzhou City (Figure 1).

Campuses of Zhejiang University are mixed communities for enrolled students, teachers, scholars, and administrative staff, with multiple activities. Taking Zijingang Campus as an example, it is the main campus of Zhejiang University, with over 70,000 people living within the campus [39] (Figure A1). Zijingang Campus currently covers an area of 3.905 km^2^ in the northeaster party of Hangzhou City, with eight gates around the campus fringe [40] (Figure A2).

It is a conventional paradigm for Chinese universities to set fences, walls, or landscape boundaries to separate from public spaces in pandemics [41]. Before COVID-19, all campuses of Zhejiang University were open to the public for free access. Someone wanting to enter the campus did not need to apply for permission but only had to pass through the entrance. There was no school policy controlling campus gates and limiting individual mobility. However, this pattern significantly changed during the COVID-19 pandemic.

### 3.2. Background of Local Pandemics

A timeline of the Hangzhou pandemic events and Zhejiang University campus operations is listed in Figure 2 to provide basic information about how the pandemic evolved in Hangzhou City and how Zhejiang University officially acted in response to COVID-19 on campus. It is shown that the first wave of the COVID-19 pandemic lasted around three months, in early 2020. There were 182 patients confirmed as infected with the virus in Hangzhou from January to March 2020. All confirmed cases were under medical segregation, away from the public. 

At that time, a significant proportion of university students had left campus and gone back home for university winter vacation. Thus, the university campus quickly closed to the public. School officials decided to delay the date of students returning to campus after winter vacation until May 2020, when the epidemic within the Zhejiang Province was eliminated. Only students living within Zhejiang Province were ordered to apply for a unique school code before returning to campus. Those who lived in other provinces had to conduct nucleic acid detection and even quarantine before entering campus. 

After the first wave, there was a loose self-policing period from July 2020 to August 2021, during which students enjoyed relatively free mobility and self-health monitoring. However, this ended when the second wave began in November 2021, when a new type of virus, the Omicron variant [42], emerged. Compared with the former conditions, it was difficult to control this wave, as confirmed infection cases in Hangzhou continued to appear for over six months after the start of the second outbreak. Notably, there was one suspected infection case in Zijingang Campus on 25 November 2021, resulting in a campus lockdown by campus officials that included self-policing (Figure A3). The officials forced students to comply with the stay-on-campus order until nucleic acid detection results of all 70,000 people were negative. Since then, the campuses have enforced strict COVID-19 restrictions, according to the Chinese dynamic zero-COVID policy [43]. The campus has not returned to being entirely open to the public, as it was before the COVID-19 pandemic, and university students’ mobility and health conditions are under daily supervision through school self-policing rules. 

### 3.3. Study Methods

#### 3.3.1. Data Collection

Social network theories suggest that users’ mental concerns and psychological well-being are reflected in social media engagement [44]. We conducted a statistical survey on Zhejiang University’s online forum CC98, where enrolled students could freely share their interests, questions, emotions, opinions, and other topics with others by posting anonymously. This social media is available only on the campus network of Zhejiang University and is free of official censorship [45], ensuring data resource accuracy for a particular focus on university students.

Usually, anonymous posting behaviors on CC98 are triggered to raise other users’ attention to group discussions on specific topics. Thus, we performed a keyword-based research strategy [46] (Table 1) on university student views to find their shifting opinions about COVID-19 and the relevant topics, particularly the campus’s self-policing management. In order to protect user privacy, the mining process at the early research stage was conducted by the forum managers. 

We conducted a diachronic research strategy to trace students’ evolutionarily changing views during COVID-19. The main timeframe was from January 2020 to August 2022, covering the entire pandemic period. Comparison data of some keywords regarding campus space use and emotional conditions were collected from January 2017 to December 2018, 3 years before the outbreak. 

#### 3.3.2. Data Analysis

This supportive dataset examined university students’ past and current concerns about pandemic situations, prevention policy, campus space use, and psychological conditions. We examined the correlations between pandemic conditions and students’ views in a monthly time-scale framework [48]. Specifically, the keywords mentioned in the post titles were counted for months as the monthly keyword frequency [49]. The higher the monthly frequency of a particular keyword, the more university students discussed it.

First, by tracing the fluctuation of monthly keyword frequency during the pandemic in the last 32 months, we aimed to investigate an overall changing trend in four aspects through 22 keywords: pandemic-oriented concerns, attitude towards restriction policies, spatial use experience, and emotions. This showed how the pandemic and relevant policies had continuously impacted university students. Second, in every aspect, by comparing monthly keyword frequency results with monthly confirmed infection cases in Hangzhou, we explored synchronous or asynchronous features of data fluctuation in the temporal dimension. In addition, comparisons among different keywords’ frequency showed which were the dominant factors. Finally, we gave speculative explanations on why data fluctuated and varied by degrees and times. 

## 4. Results

### 4.1. What Were University Students’ Concerns Related to the Pandemic?

Figure 3 illustrates the add-up monthly keyword frequency of “Epidemic (疫情)”, “COVID-19 (新冠)” and “Omicron (奥密克戎)” that appeared in the post titles on CC98. The number of confirmed infection cases in Hangzhou was used to evaluate the local pandemic situation. Several features could be summarized, as follows: University students talked about the COVID-19 pandemic on the university forum over the past 32 months;Though there were no new cases of confirmed infection in Hangzhou from April 2020 to October 2021, students still cared about the pandemic situations in the other cities, their hometowns, and related destinations;The frequency peak month was April 2022, when a nearby megacity—Shanghai—fell into a severe breakout, while no case was found in Hangzhou [50]. A total of 395 posts were created in this month, when panicked students discussed the accelerating epidemic crisis in Shanghai. They posted worries about the potential risks of pandemics spreading from Shanghai to Hangzhou. After Shanghai ended its lockdown in June, the keyword frequency dropped accordingly.

### 4.2. How University Students Reacted to the Epidemic Prevention Policies from the Hangzhou Government and Zhejiang University

Figure 4 shows a periodic fluctuation of keyword frequency about the pandemic prevention policies issued by the local government and Zhejiang University. Posts on six main policy-related topics (“Health code”, “School code”, “Daily health information update”, “Nucleic acid detection”, “Prevention policy”, and “Quarantine”) became gradually frequent when it was close to the month of the academic semester’s start and end. The first peak period of keyword frequency was from May to July 2020, when students continuously returned to campus. This was followed by a winter vacation in January 2021 and January 2022, and a summer vacation in August 2022. Orders restricted students’ mobility between campus and public destinations like hometowns and internship companies.

However, during the second pandemic wave, relevant concerns and complaints about epidemic prevention policies increased, even in the middle of a semester. This reflected an impact on daily mobility. Students gradually showed intensive concerns about when and how to end restriction orders and campus lockdown management as, since April 2022, there were no new confirmed infection cases in Hangzhou. 

Figure 5 gives details about university students’ specific concerns about campus access. Since the school strengthened gated entrance control after the first outbreak, students were deprived of the freedom to enter campus whenever they wanted. It was thought that if students entered the campus from other cities, there might be a chance to carry the virus into campus. Thus, the periodical peak of keyword frequency “enter campus (进校)” appeared in the month before the spring or autumn semester started, usually in February and August. In comparison, the keywords “leave campus” were less frequently mentioned by students, and the school did not impose restrictions on students’ behaviors outside the campus.

It took several months for university students to accept a new policy and obey top-down orders for epidemic restrictions that limited students’ traveling and academic activities (Figure 6). The first invention of the school code raised broad discussion on the forum in mid-2020. They were sensitive to the orders from the school authority that they had to obey, especially when they violated their freedoms. Then, the focus on school code regulations gradually declined; most students increasingly accepted them as time passed. A similar fluctuation was reflected in the monthly keyword frequency of “Nucleic acid detection”, which had a sharp frequency peak in January 2021 and then fell in the following months. This was the first time the government and the school ordered students to perform regular detection before returning home or entering campus.

During the second wave, epidemic prevention policies changed rapidly. The restrictions were sometimes loose, and at other times, tight. Rising keyword frequency reflected students’ accumulated impatience with repetitive and endless prevention policies. In recent months, there has been an increasing number of posts on CC98 talking about the necessity of nucleic acid detection (Figure A4), which students are ordered to conduct every three days (or sometimes seven days). If they do not comply with this duty, they lose the chance to enter campus again. Some online complaints said such frequent nucleic acid detection was an inconvenience to daily routines. Others questioned the efficiency and benefits of obeying these complex COVID-19 prevention policies. 

### 4.3. How Did University Students’ Use of Campus Spaces Change in a Lockdown Campus Environment during COVID-19?

Figure 7 gives information about the monthly keyword frequency of campus space use since 2017. Data from the earlier three years are used to compare to the changes during the pandemic. Generally, university students’ concerns about space use have increased obviously. Their focus on the living spaces of dormitories and canteens (Figure A5) has grown rapidly due to the decreased chance to sleep or eat off campus. Most posts were full of negative comments on the services, prices, and quality of the campus living environment and food, which became outlets for expressing dissatisfaction with the authorities. 

In contrast, the monthly keyword frequency of study space use fluctuated modesty but similarly showed an increasing trend. We aimed to provide speculative explanations for such a spatial discussion from several viewpoints, based on the on-site observations and feedback from interviewers:The campus generally had a tight bond with the public neighborhoods before the pandemic, where urban public facilities and infrastructure like shopping malls, restaurants, cinemas, etc., enriched the on-campus life of university students.After campus lockdown and self-policing management, campus living spaces and services failed to meet the diverse campus life needs of hundreds of thousands of students, especially in eating and entertaining.One reason for the lower keyword frequency of studying spaces than living spaces on the school forum was that the widely used remote classrooms relieved academic loads on campus hardware facilities, which partially compromised the geographic barriers put in place by the campus mobility restrictions.

Figure 8 further shows the online academic conditions of university students. Due to the pandemic’s negative impacts on academic connections offline, there are rising channels of online classes, online lectures, and remote internships as replacements. In the beginning, Zhejiang University quickly introduced online classrooms for learning and teaching as an emergent solution for the delay of campus opening after the breakout in early 2020. 

However, such an online teaching and learning model continued after students returned to campus in May 2020. The regular teaching schedule was transformed into a remote model for half a year, until September. With the increasing advantages of online academic activities, university students kept open minds to sharing online academic resources and recruitment information on the forum with others. This contributed to self-learning freedoms, where students considered which online classes to take and how they could benefit from others’ experiences with online classes. 

According to the monthly keyword frequency, remote internships and practices received less attention than online classes. The latter was static and single-acting, as teachers pre-recorded their lectures for the student audience. However, internships and laboratory experiments required higher interaction and real-time feedback among different groups. Thus, their advantages through online channels have not been fully developed, with a few students mentioning them on CC98.

### 4.4. How Have University Students’ Emotional Conditions Changed in a Lockdown Campus Environment during COVID-19?

Figure 9 shows the mental health–oriented keyword frequency fluctuations of university students. “Happiness (开心)” and “Depression (抑郁)” respectively represented positive and negative expressions of students’ emotions in the online posts. Moreover, “Complaint (吐槽)” and “Insomnia (失眠)” were two research keywords used to describe emotional behaviors.

The monthly “Happiness” frequency experienced a sharp increase from an average of fewer than 40 posts in a month before COVID-19 to nearly reaching 138 posts in January 2022. This shows an interesting trend for university students to express their happiness to other anonymous users on the forum during the pandemic. Especially during times such as the Chinese Spring Festival and graduation seasons, the online forum was full of greetings and congratulations from unknown students. Such an communication was not seen regularly before the pandemic.The monthly “Depression” frequency had a slow rise until August 2021, then started to jump in the following months. It peaked in March 2022, when university student users created 127 posts expressing painful, distressed, or depressed feelings.The monthly “Complaint” frequency fluctuated over an academic semester in cycles. The average number of complaints in a semester went up modestly.The monthly “Insomnia” frequency surged during COVID-19. Before the pandemic, a few students (1–3 posts monthly) talked about their sleeping problems with the others on CC98. However, we found more and more posts that mentioned poor sleeping conditions posted at midnight. This phenomenon arose in April 2022.

## 5. Discussion

### 5.1. Reflections on the Self-Policing Management on the University Campus

This study provides evidence from students’ online posts at Zhejiang University to examine the benefits and costs of self-policing management on campus. The benefits were obvious: there were no infection cases in high-density-population campuses during the pandemic. However, this does not mean that the current self-policing mode would be the optimal decision in the future. Analysis of the keyword frequency of online forum post titles reflects students’ responses to self-policing and reveals the multiple impacts on university students during COVID-19.

We systematically illustrate the logic of current self-policing management on the Zhejiang University campus (Figure 10). To prevent the pandemic, university officials enacted additional policies to restrict students’ mobility in addition to public prevention policies ordered by the local government. 

Whenever a new COVID-19 prevention policy was enacted, it had the potential to lead to significant impacts on young university students who had suffered from mood swings during the pandemic for over 32 months. The trend toward spatial privatization was under the pretext of COVID-19 prevention, constraining university students’ freedom of social mobility. The school officials’ dominant behaviors of campus enclosure and self-policing management had the potential to lead to hegemonic institutions on university campuses, with little open communication between the students and the officials. 

The lockdown actions toward emergent pandemics on campus are intended to be temporary and adapt to flexible circumstances rather than impose rigid movement control. On the other hand, they should function without overstepping students’ acceptance. Overwise, ongoing intervention from university officials poses a potential crisis in advanced education in the future if such a self-policing mode is fixed as a common campus rule. 

The impacts of prevention policies are more profound than the pandemic itself. To date, no student at Zhejiang University reported virus infection; however, almost all students were involved in the school restriction mechanism. This leads to consideration of so-called “right” decisions and “effective” strategies for pandemics as implemented in the past and currently. A warning signal is students’ online forum topic keyword frequency in light of the current self-policing management on the university campus, whose benefits and costs were beyond the normal routines of university life. 

Comparing the second pandemic wave with the first, we found that university students posted many more resentful opinions and feelings about the pandemic situation, school policies, and spatial uses. This reflected a change from obedient to rebellious in the second wave, from September 2021 until now. They did not wholly acclimatize themselves to the top-down restrictions from the government and school officials. Therefore, mainly due to the persistent Omicron prevention, innovation is needed rather than restriction.

### 5.2. An Introverted Trend under the Circumstance of Self-Policing on Campus 

The university students’ online postings reveal an introverted trend in their character, lifestyle, interests, and behaviors. Since the school authority restricted students’ daily mobility between the campus and the public and reduced collective actions like parties, it unconsciously promoted staying inside rather than frequent individual–society interactions. Campus enclosure led to social and emotional isolation, where students’ attention partially moved from external factors toward themselves. 

The Internet was an important channel for the students to gain access to others without violating the pandemic restriction policies. In the virtual world, social connection and mobility are not geographically interrupted. Students on CC98 shared their thoughts with strangers to gain instant attention and support without any need for social distance. They built a close relationship with people in a similar circumstance through online chatting. For example, when a student could not enter the campus without a school code, he or she would win sympathy from others by posting. These emotional benefits were used for real-world deficiencies. Usually, a post has limited time efficiency, with new posts quickly replacing older ones. Despite this, users still enjoyed a sense of affirmation in a short period. 

However, these online social activities happened individually, magnifying university students’ introverted feelings in their inner worlds. This led to so-called networked individualism [51].

### 5.3. Study Limitations and Future Plans

There are several limitations in the study that should be clarified. First, this study used CC98 data to monitor university students’ concerns by collecting the keyword frequency of post titles. However, forum users cannot represent the overall features of entire university groups. There are numerous students and scholars who are not familiar with CC98. Second, there are some inevitable factors influencing posting behaviors. For example, during winter vacation in February and summer vacation in July and August, students who are away from campus have a lower chance of using CC98. This is an objective factor in the periodic decrease of keyword frequency. Third, the keyword is not equal to semantics recognition; thus, calculating keyword frequency could not wholly represent the actual orientations of the posts.

Such self-policing management is prevalent in other environments, including in residential communities, nursing homes, primary schools, and middle schools. Comparatively speaking, a university campus is a particular type of living neighborhood for self-policing, where social activities are relatively simple. Nevertheless, in other places, for example, in mixed neighborhoods with locals and urban–rural migrants, community managers might use different approaches to restrict individual mobility and reduce the risk of an outbreak. 

Connecting university students with other social groups (younger generations, low-income individuals, ethnic groups, etc.), whether under similar or different conditions, deserves further attention. For example, Liu et al. [52] revealed a rising social concern of disadvantaged populations who were excluded and marginalized from public smartphone-based services during the COVID-19 pandemic. Though university students partially relieved the pandemic impacts of campus lockdown by going online, others who are less technology-savvy might suffer more inequality and inconvenience in their communities and public places. 

## 6. Conclusions

Universities have made great efforts in pandemic prevention through self-policing to keep COVID-19 away from student agglomerations. However, in response to the Omicron wave, a resilient and inclusive transition is needed to improve the current policy’s declining benefits and rising disadvantages. This study revalues the impacts of self-policing at Zhejiang University by examining young students’ online posts, which recorded their thinking and mental fluctuations during the pandemic. University students became sensitive to how the pandemic developed and how the local government and university officials exercised their power to take preventive measures. Compared with the keyword frequency about the pandemic situation, that regarding epidemic prevention policies was higher. University students paid more attention to their direct interests on campus and worried about their limited freedom; in turn, their relations with and trust in authorities declined.

In a long-term self-policing campus environment, university students’ feedback provides potential improvement to campus management in the future. We cannot give specific predictions or suggestions on when and how to return to a balanced and resilient campus morphology. However, this study may lead to further consideration of the school officials’ self-policing policies and the related power structures. Since university students under suppression cannot achieve all-around personal development, the cause of students’ resentment needs to be addressed [53]. 

## Figures and Tables

**Figure 1 ijerph-19-12172-f001:**
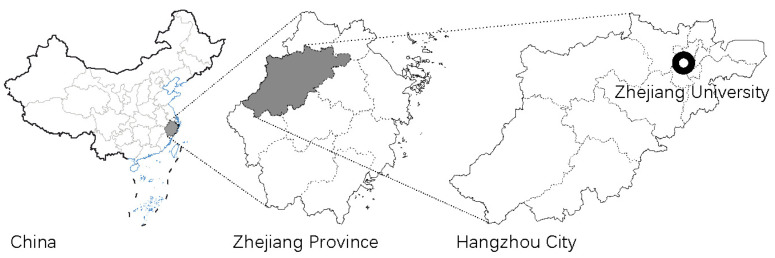
Zhejiang University, Hangzhou City, Zhejiang Province, China.

**Figure 2 ijerph-19-12172-f002:**
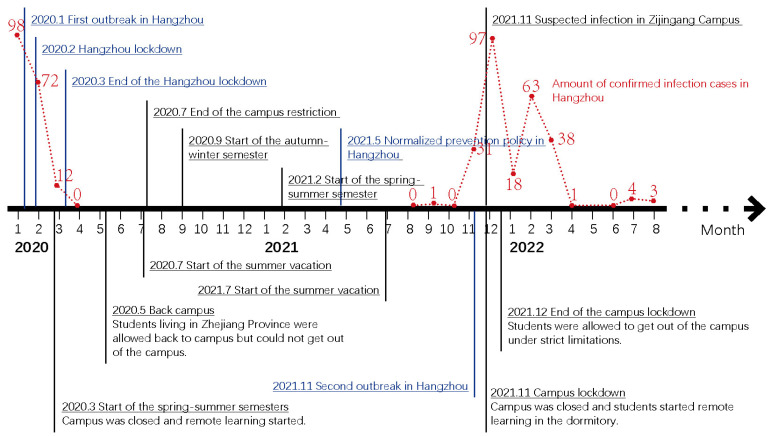
Timeline of pandemic events in Hangzhou and the corresponding campus operations of Zhejiang University.

**Figure 3 ijerph-19-12172-f003:**
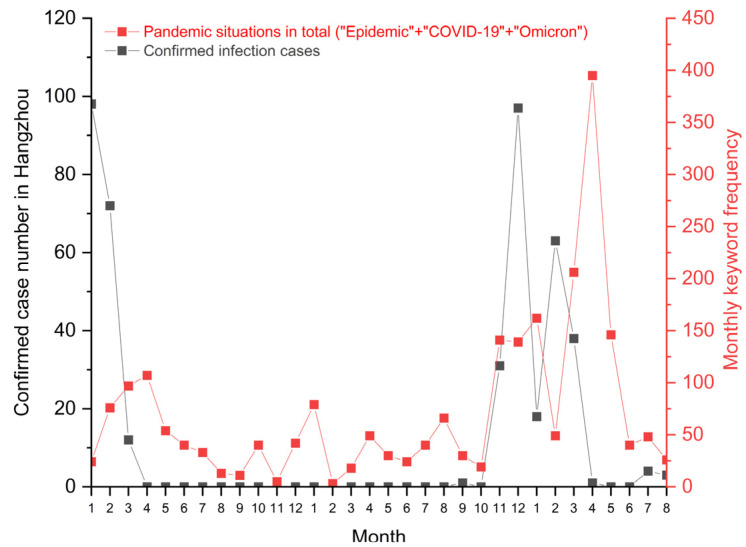
Monthly keyword frequency about the pandemic as compared with the number of confirmed infection cases in Hangzhou.

**Figure 4 ijerph-19-12172-f004:**
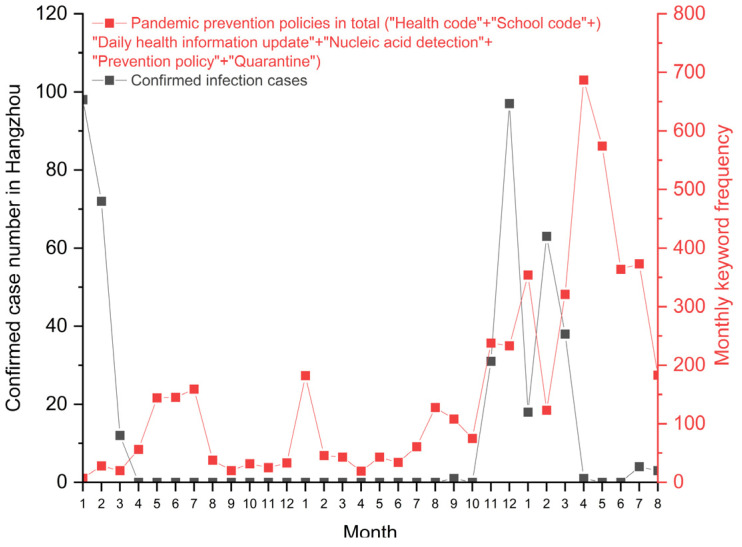
Monthly keyword frequency about epidemic prevention policies as compared with the number of confirmed infection cases in Hangzhou.

**Figure 5 ijerph-19-12172-f005:**
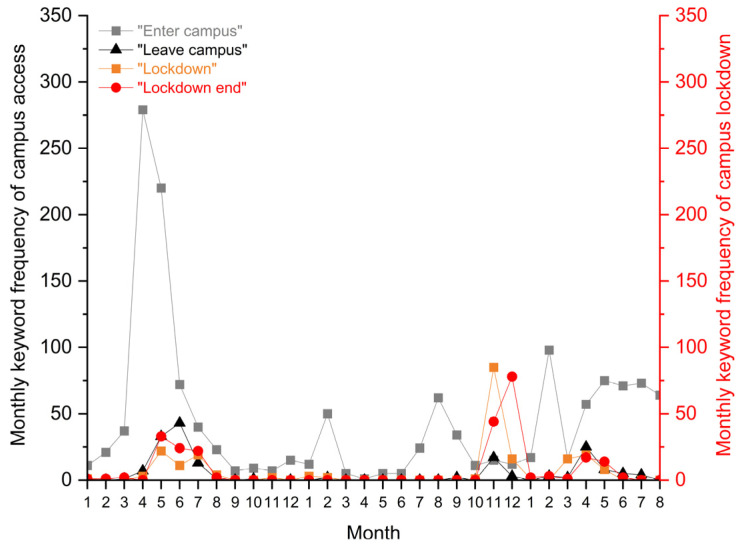
Monthly keyword frequency about campus access control: “Enter campus”, “Leave campus”, “lockdown”, and “lockdown end”.

**Figure 6 ijerph-19-12172-f006:**
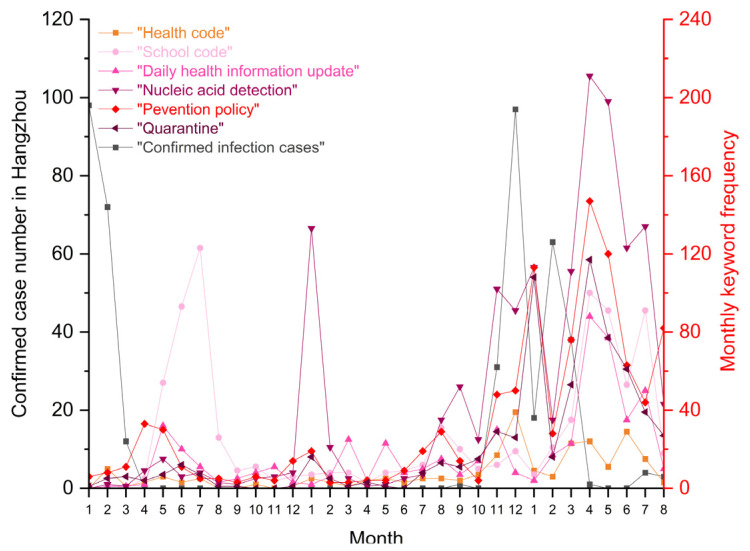
Monthly keyword frequency: “Health code”, “School code”, “Daily health information update”, “Nucleic acid detection”, “Prevention policy”, and “Quarantine”.

**Figure 7 ijerph-19-12172-f007:**
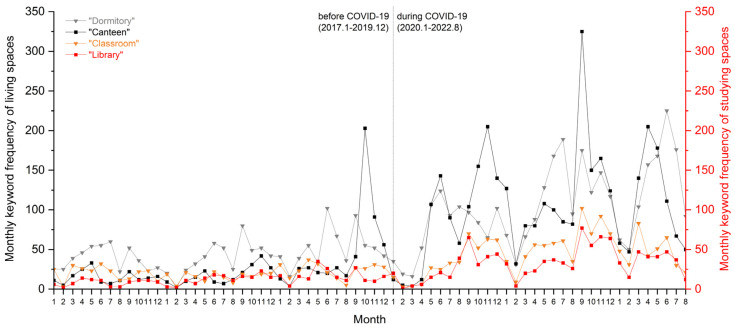
Monthly keyword frequency: “Dormitory”, “Canteen”, “Classroom”, and “Library”.

**Figure 8 ijerph-19-12172-f008:**
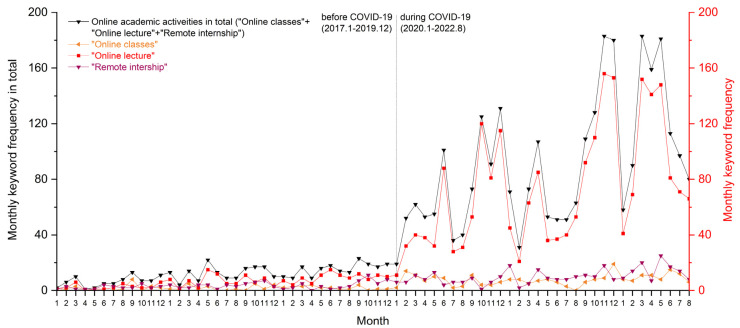
Monthly online-related keyword frequency: “Online classes”, “Online lecture”, and “Remote internship”.

**Figure 9 ijerph-19-12172-f009:**
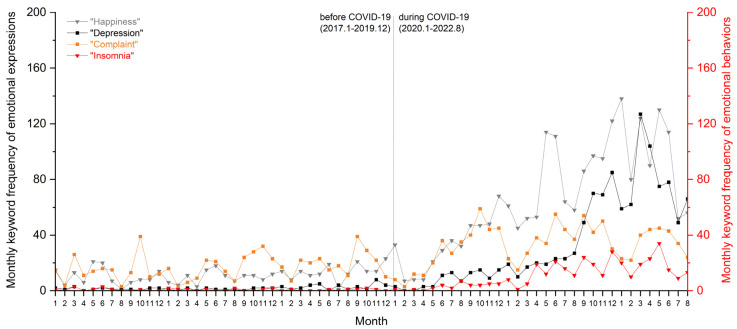
Monthly keyword frequency about emotional conditions: “Happiness”, “Depression”, “Complaint”, and “Insomnia”.

**Figure 10 ijerph-19-12172-f010:**
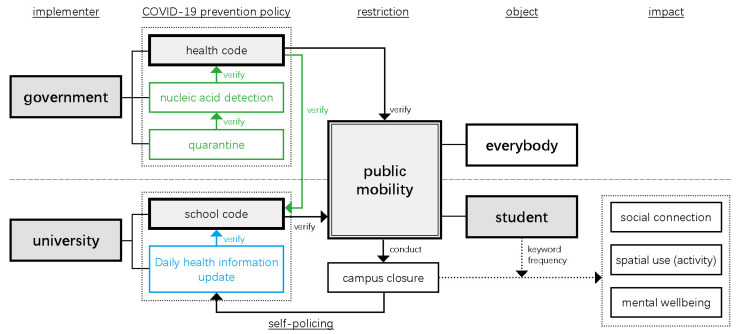
Illustration of current self-policing management on campus.

**Table 1 ijerph-19-12172-t001:** A research list of Zhejiang University students’ online discussion topics and keywords in their post titles on CC98.

Discussion Topics	Keywords in the Post Titles
Pandemic situations	“Epidemic (疫情)”
	“COVID-19 (新冠)”
	“Omicron (奥密克戎)”
Epidemic prevention policies	“Health code (健康码)” ^1^
	“School code (蓝码)” ^2^
	“Daily health information update (打卡)” ^3^
	“Nucleic acid detection (核酸检测)”
	“Prevention policy (防疫政策)”
	“Quarantine (隔离)” ^4^
Campus access control	“Enter campus (进校)”
	“Leave campus (离校)”
	“Lockdown (封校)” ^5^
	“End lockdown (解封)”
Campus space use ^6^	“Dormitory (寝室)”
	“Canteen (食堂)”
	“Classroom (教室)”
	“Library (图书馆)”
	“Online” ^7^
Emotional conditions	“Happiness (开心)” ^8^
	“Depression (抑郁)” ^9^
	“Complaint (吐槽)”
	“Insomnia (失眠)”

^1^ “Health code (健康码)” is a public electronic code commonly used in China, which records individual risk of infection. It has three colors: green, yellow, and red, representing low risk, medium risk, and high risk, respectively. The health code is automatically related to the latest results of nucleic acid detection. ^2^ “School code (蓝码)” is an electronic code in a particular mobile app invented by Zhejiang University for every enrolled student. It has two colors: blue and grey. Zhejiang university issued a campus access policy that only someone whose unique school code is blue is eligible to enter the campus. The rules of obtaining a unique blue code are constantly changing in accordance with the local pandemic situation. ^3^ Zhejiang University orders every enrolled student and staff to update their daily health information through a specific online platform. ^4^ Quarantine is a strategy applied to university students before entering campus from high infection-risk areas [47]. ^5^ The university campus is walled and enclosed with strict identification checks of every visitor’s school code by campus guards. ^6^ Regarding the campus spaces, we categorized them into two functional types: living spaces (dormitory and canteen) and studying spaces (classroom and library). These four kinds of campus spaces represent where university students spend most of their daily life without leaving the campus. ^7^ “Online ” is used as a synonym for three keywords in Chinese: “网课”, “线上讲座”, and “远程实习”. ^8^ “Happiness ” is used as a synonym for three keywords in Chinese: “开心”, “快乐”, and “幸福”. ^9^ “Depression ” is used as a synonym for five keywords in Chinese: “抑郁”, “自闭”, “痛苦”, “emo”, and “难过”.

## Data Availability

Not applicable.

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
