# Peer review of "Multi-Impacts of Spatial Self-Policing during COVID-19: Evidence from a Chinese University"

_ijerph, 2022, doi:10.3390/ijerph191912172_

Round 1
Reviewer 1 Report
Comments and suggestions are in attached file.

Reviewer 2 Report
Thanks for inviting me to review this manuscript. This paper investigates the plights of university students during the pandemic in China. I have to say I like this paper. Although many people have already investigated university students’ everyday lives, their mental wellbeing, and their physical activities in a wide range of contexts, previous papers (I read and reviewed) had not touched upon the interconnectedness between covid policies and those experiences. That is exactly what I repeatedly argued in the past two years—it was the improper containment policy that caused social problems, not the virus itself. In this sense, I am quite excited to see those very interesting details the authors provided in this paper. Thank you for sharing.
But, I believe this paper can still be improved in the following ways:
1. In the methodology section, particularly section 3.2.2, the authors did not make it very clear how you analyse the data. It seems that the authors conducted some sort of big text data analysis, but what exactly it is? What analyses you actually did? This has to be clarified in section 3.2.2.
2. It is some sort of longitudinal study (almost three years, that’s an invaluable dataset!), or maybe it uses panel data. Then it’s better to show how their attitudes of various sorts have changed over time. It must be most interesting to see how (and maybe when) they changed from obedient to rebellious.
3. Since Covid-19 increased censorship in China (Chang et al., 2022) and many words may be perceived as sensitive by students. They may not use the exact word such as “pandemic”. For example, I have been locked in my home in Shanghai for three months, we use “sheep” for those who tested positive. I think you can add this as one of your limitations in the discussion section.
4. I would like to see an additional section in the discussion section, connecting university students with other socially disadvantaged populations. Because they have been indeed another extremely disadvantaged but unfortunately overlooked group during the pandemic. Perhaps because the group of university students is politically sensitive as it has been related to so many political taboos. I am a bit confusing, because Liu et al. (2021) found that younger population groups have a much higher level of accessibility of various activities because those activities can be easily accessed via smartphone whilst many other less technology-savvy people cannot do. Are these conflicting? Another thing, in Liu et al. (2022)’s study on rural-urban migrants in the YRD region, they found that migrants originally from different places may be enemies of other migrants. Migrants from certain provinces suffered from discrimination from other migrants and consequently making the living environment extremely unpleasant. Have you found anything like this? Were those students sharing the same view without any disagreement?
Again, this is a very interesting paper and the sounds of university students should be heard.
Reference
Chang, K. C., Hobbs, W. R., Roberts, M. E., & Steinert-Threlkeld, Z. C. (2022). COVID-19 increased censorship circumvention and access to sensitive topics in China. Proceedings of the National Academy of Sciences, 119(4), e2102818119.
Liu, Q., An, Z., Liu, Y., Ying, W., & Zhao, P. (2021). Smartphone-based services, perceived accessibility, and transport inequity during the COVID-19 pandemic: A cross-lagged panel study. Transportation Research Part D: Transport and Environment, 97, 102941.
Liu, Q., Liu, Z., Kang, T., Zhu, L., & Zhao, P. (2022). Transport inequities through the lens of environmental racism: rural-urban migrants under Covid-19. Transport policy, 122, 26-38.
Round 2
Reviewer 2 Report
Thanks for addressing my previous comments. I am generally satisfied with this manuscript.
A few typos:
line 169 a pandemic is an epidemic that spreads over countries and continents, so "local pandemic" sounds very weird. maybe use "epidemic" instead. an epidemic is a disease that affects many people within a community, so perhaps this is a better word here.
line 744 I don't understand who are the "crowds", in case it means laobaixing in Chinese, it's proper to use "lay-citizens" here.
line 751 "taking an active part online" sounds non-English, it could be "participating in activities online" or maybe "taking an active part in online activities"
please double check
